# A Novel Virtual Optical Image Encryption Scheme Created by Combining Chaotic S-Box with Double Random Phase Encoding

**DOI:** 10.3390/s22145325

**Published:** 2022-07-16

**Authors:** Peiran Tian, Runzhou Su

**Affiliations:** Department of Physics, College of Science, Northeast Forestry University, Harbin 150040, China

**Keywords:** chaotic map, substitution box, virtual optical image encryption, DRPE

## Abstract

The double random phase encoding (DRPE) system plays a significant role in encrypted systems. However, it is a linear system that leads to security holes in encrypted systems. To tackle this issue, this paper proposes a novel optical image encryption scheme that combines a chaotic S-box, DRPE, and an improved Arnold transformation (IAT). In particular, the encryption scheme designs a chaotic S-box to substitute an image. The chaotic S-box has the characteristics of high nonlinearity and low differential uniformity and is then introduced to enhance the security of the DRPE system. Chaotic S-boxes are resistant to algebraic attacks. An IAT is used to scramble an image encoded by the DRPE system. Meanwhile, three chaotic sequences are obtained by a nonlinear chaotic map in the proposed encryption scheme. One of them is used for XOR operation, and the other two chaotic sequences are explored to generate two random masks in the DRPE system. Simulation results and performance analysis show that the proposed encryption scheme is efficient and secure.

## 1. Introduction

With the rapid development of computers and the Internet, information security has received extensive attention from academia and industry, in which the security protection of image data is a significant and current research topic. To guarantee the security of images, researchers have developed many techniques including image encryption [1,2,3], watermarking [4,5], and data hiding [6,7], wherein image encryption is the most direct and effective method. Owing to the characteristics of high speed, and multi-dimensional and parallel processing of optical systems, the researcher took a keen interest in the study of optical image encryption [8,9]. For example, the double random phase encoding (DRPE) architecture based on a 4f optical system was first proposed by Refregier and Javidi [10]. 

DRPE can encrypt images into stationary white noise. To achieve better encryption performance, many image encryption schemes combining DRPE with other encryption methods such as fractional Fourier transform [11,12], Fresnel transform [13,14], and gyrator transform [15,16] were proposed. 

Although DRPE is an effective scheme, it is a linear encryption process; as a result, DRPE-based image encryption algorithms are vulnerable to some specific attacks. To tackle this issue, many scholars have recently studied nonlinear optical image encryption. For instance, Qin et al. proposed a secure nonlinear cryptosystem based on phase truncation in the Fourier transform domain, which overcame the linear weakness of DRPE systems [17], but it was cracked by the authors of [18]. Chen et al. proposed a method to enhance the security of DRPE using phase preservation and compression and applied a nonlinear correlation algorithm to authenticate the decrypted image to improve the security of the DRPE system [19]. Wang et al. proposed an encryption-efficient asymmetric optical image encryption algorithm based on improved amplitude and phase recovery [20]. By introducing nonlinear terms, Dou et al. developed a novel DRPE system [21]; it was, however, cracked by the authors of [22]. In addition, Faragallah et al. introduced an efficient compression encryption scheme based on research results of the SPIHT/JPEG compression of DRPE-encrypted images based on optical discrete cosine transform (DCT), where the compression was involved to reduce the bandwidth requirement in the communications [23].

Since chaotic systems have inherent properties of ergodicity, pseudo-randomness, and sensitivity to initial conditions and control parameters, many chaos-based cryptosystems were proposed in the past decade [24,25,26,27,28]. For example, Li et al. proposed to utilize the Arnold transform to scramble the pixels in the local area of the complex function composed of two original images, and then conducted the gyrator transform operation to encrypt the scrambled image [24]. Sui et al. developed a double-image encryption algorithm with discrete fractional random transformation and logistic mapping. In the proposed algorithm, the logistic mapping was applied to the diffusion process such that the correlation between adjacent bit planes could be reduced [25]. Lang et al. designed an asymmetric optical color image encryption algorithm using compressed sensing and quantum logistic mapping. In this algorithm, compressed sensing was explored to reduce data transmission, and quantum chaotic mapping was utilized to generate random phase masks [26]. Huang et al. designed a nonlinear multi-image encryption algorithm with a chaotic system and two-dimensional (2-D) linear regular transformation. This algorithm could encrypt images in the space domain and frequency domain and compress multiple images into a small amount of data [27]. Additionally, Abd El Atty et al. developed a novel optical encryption method using the quantum walk and a DRPE system, where the quantum walk was applied for generating the two random masks used in the DRPE system and the permutation and diffusion of an original image as well [28]. From the above research results, it can be seen that the application of chaotic systems in cryptography is in the ascendant.

In encryption systems, an S-box is a crucial module that plays a significant role in confusion and diffusion in block cipher systems such as Advanced Encryption Standard (AES) and Data Encryption Standard (DES). An *n* × *n* S-box is a nonlinear map S: {0, 1}n → {0, 1}n, with {0, 1}n representing the vector spaces of n elements from GF(2) [29,30]. Traditional cryptography utilizes an algebraic method to construct an S-box to achieve a high degree of nonlinearity. However, the constructed S-box has weak differential performance and is vulnerable to algebraic attacks [31,32]. To tackle this issue, the researcher constructed many chaotic S-boxes [33,34,35,36,37,38,39,40,41,42,43,44,45,46,47,48,49,50,51,52,53,54,55]. In 2001, Jakimoski et al. first constructed an S-box based on discrete chaotic systems [33]. Since then, various chaotic S-box construction methods were proposed based on discrete-time chaotic systems [34,35,36,43,45,50,53,54,55], continuous-time chaotic systems [37,38,39,40,41,42,44], hyper chaotic systems [46,47], and joint optimization and chaotic systems [48,49,51,52]. The computational complexity of optimally generating S-boxes is highest, followed by high-dimensional continuous-time chaotic systems, and low-dimensional discrete chaotic systems is the lowest. Therefore, the generation of S-boxes for low-dimensional discrete chaotic systems has received extensive attention.

Recently, some image encryption algorithms using chaotic S-Boxes have been proposed [56,57,58,59,60,61,62,63]. For example, Zhang et al. developed an efficient chaotic image encryption scheme using alternate circular S-boxes with the chaotic S-boxes being produced by the Chen chaotic system [56]. Belazi et al. introduced a novel image encryption scheme based on a substitution-permutation structure and a chaotic S-box constructed by the logistic-sine map [57]. Lu et al. proposed an image encryption algorithm using a chaotic S-box generated by the discrete compound chaotic map [58]. The performance analysis showed that the proposed encryption algorithm exhibited outstanding diffusion and confusion properties. Idrees et al. proposed a cryptosystem using a substitution-permutation network, where the S-box was constructed by the chaotic sine map [59]. Wang et al. developed a novel image encryption scheme employing a spatiotemporal chaos-based dynamic S-box and random blocks [60]. This S-box was constructed using one-dimensional (1-D) and 2-D chaos mapping and DNA sequencing [61]. Simulation results showed that the proposed algorithm had the ability to resist many kinds of attacks. Ali et al. utilized the generated S-box by PWLCM, the tent logistic system, to design a novel image encryption scheme [62]. In addition, Deb et al. proposed a new algorithm with key-dependent bijective S-Boxes under a 2-D Henon-map [63]. The performance analysis showed that this algorithm achieved good confusion and diffusion capabilities. Therefore, encrypting images based on chaotic S-boxes can produce good encryption effects.

S-boxes were introduced in some algorithms to enhance DRPE [64,65,66]. Hussain et al. proposed to utilize an information-hiding technique to design an optical image encryption algorithm [64]. In the designed algorithm, the DRPE and S-box transformation were applied. They also designed an optical image encryption system based on a fractional Hartley transform and an S-box using linear fractional transform and chaotic maps [65]. Girija et al. proposed to encrypt images with DRPE and a random S-Box [66]. However, the research on the combination of DRPE and the Chaotic S-box to achieve a high degree of nonlinearity has been under-studied.

The main contributions of this work are summarized as follows:(1)Exploring a chaotic map and conducting the transformation of stretch and fold to construct an efficient and secure S-box. The cryptographic performance of the constructed S-box is testified.(2)A secure image encryption scheme is developed by integrating the chaotic S-box, DRPE, and IAT. In this scheme, the plaintext information is involved in the control parameters of IAT. All the gray pixel values are substituted by the S-box. We XOR the substituted image with a nonlinear chaotic sequence. Next, the two random-phase masks are generated by a nonlinear chaotic map. The XOR result is encoded by DRPE, and the obtained result is further confused by IAT.(3)Simulation and security analysis are conducted to verify the effectiveness of the proposed encryption scheme. Simulation results and performance analysis show that the proposed scheme is efficient and secure.

The remainder of the present paper is organized as follows. Some relevant fundamental knowledge is briefly introduced in Section 2. Section 3 presents the S-box obtained in our paper. The proposed cryptosystem approach for grayscale images is described in Section 4. Section 5 shows results and security analyses. Finally, we summarize this work in Section 6.

## 2. Fundamental Knowledge

### 2.1. Overview of Optical DRPE Cryptosystem

The following encryption processes are performed on the original image: (1) modulate the original image with the first random phase mask placed in front of the input plane; (2) modulate the obtained result with the second random phase mask placed in front of the output plane to obtain an encrypted image [9]. After that, the encrypted image becomes stationary white noise of complex amplitude.

Figure 1 depicts the principle of the optical DRPE cryptosystem. The process of encryption and decryption is given by Equations (1) and (2):(1)$$G(x,y)=FT^{-1}\{FT\{F(x,y)\cdot \exp[i2\pi m(x,y)]\cdot \exp[i2\pi n(u,v)]\}\}$$
(2)$$F(x,y)=FT^{-1}\{FT\{G(x,y)\cdot \exp[-i2\pi n(u,v)]\cdot \exp[-i2\pi m(x,y)]\}\}$$
where *F*(*x*, *y*) and *G*(*x*, *y*) denote plain and cipher images, respectively; FT{·} indicates the *FT*, and *FT*^−1^ {·} represents the IFT. Two random phase masks m(x,y) and n(u,v) with exp[i2πm(x,y)] and exp[i2πn(u,v)] are uniformly distributed between 0 and 1.

### 2.2. Nonlinear Chaotic Map

The formulas of a nonlinear chaotic map are given by Equation (3) [67]:(3)$$\begin{array}{l} x_{n+1}=[\mu \times k_{1}\times {\left(1-x_{n}\right)}^{2}\times y_{n}+z_{n}]\mathrm{mod}1\\ y_{n+1}=[\mu \times k_{2}\times x_{n+1}+(1+{\left(y_{n}\right)}^{2})\times z_{n}]\mathrm{mod}1\\ z_{n+1}=[\mu \times k_{3}\times \sin(x_{n+1})\times z_{n}+y_{n+1}]\mathrm{mod}1 \end{array}$$
where 𝜇, 𝑘_𝑖_ are system parameters of the nonlinear chaotic map and when 0 < 𝜇 ≤ 3.999, |𝑘_1_| > 29.5, |𝑘_2_| > 31.3, |𝑘_3_| > 25.1. With 𝑘_1_ = 32.5, 𝑘_2_ = 34.3, 𝑘_3_ = 28.1, and 𝜇 ∈(0,3.999], this system exhibits chaotic bifurcation diagrams and Lyapunov exponents as shown in Figure 2a–d, respectively. When 𝑘_1_ = 32.5, 𝑘_2_ = 34.3, 𝑘_3_ = 28.1, and 𝜇 = 3.999, the chaotic attractor of this system is shown in Figure 2e.

### 2.3. Improved Arnold Transformation

In the 1960s, Vladimir Arnold discovered the Arnold map (AM), which has no attractor and is used to scramble images [68].

AT can be described as the following Equation (4):(4)$$\left(\begin{array}{l} x_{n+1}\\ y_{n+1} \end{array}\right)=\left(\begin{array}{l} a\;b\\ c\;d \end{array}\right)\left(\begin{array}{l} x_{n}\\ y_{n} \end{array}\right)\mathrm{mod}N$$
where the four parameters *a*, *b*, *c*, and *d* are all positive integers in AT, and gcd(*ad* − *bc*, *N*) = 1.

The Equation (4) can be further transformed as follows:(5)$$\begin{array}{l} x_{n+1}=(ax_{n}+by_{n})\;\mathrm{mod}\;N\\ y_{n+1}=(cx_{n}+dy_{n})\;\mathrm{mod}\;N \end{array}$$

To improve the performance of AT, we should use Equation (6) rather than Equation (5) [69].
(6)$$\begin{array}{l} x_{n+1}=(x_{n}+hy_{n})\;\mathrm{mod}\;N\\ y_{n+1}=(gx_{n}+(hg+1)y_{n}+e(x_{n+1}^{r}+k))\;\mathrm{mod}\;N \end{array}$$
where r∈{2,3,4,5,6,7} and k∈{1,2,…,255}.

The nonlinear term makes the improved AT change the deficiency of quasi-affine transform and enhances the ability and security of AT against differential attacks.

### 2.4. Transform of Stretch and Fold

a.The stretch transform of nonadjacent rows and columns

The procedure of stretch transform of nonadjacent rows and columns [70] is as follows: insert the pixels of a row or column of the image pixel matrix into other adjacent rows or columns of pixels, stretch the obtained pixels into 1-D series, and then fold them to the same size as the original image matrix. This will guarantee efficient scrambling of the original adjacent pixels (not at the original positions). Figure 3 and Figure 4a,b illustrate the detailed process.

b.The fold transform of a snake line

The process of fold transform of a snake line can be described as the following: arrange the pixels in the image matrix in an order of a snake line, put the obtained pixels into 1-D series, and then fold them to the same size as the original image matrix [70]. Figure 3 and Figure 4c show the detailed process.

## 3. S-Box Construction and Evaluation Criteria

### 3.1. The Proposed S-Box Generation Scheme

**Step****1:** We iterate the chaotic system Equation (3) with initial values ${x^{\prime }}_{0}$, ${y^{\prime }}_{0}$, ${z^{\prime }}_{0}$. For the *N*_0_th iteration,${x^{\prime }}_{1}$, ${y^{\prime }}_{1}$, ${z^{\prime }}_{1}$ can be obtained. Then, *S* is defined as an array of 256 integers.

**Step****2:** We take ${x^{\prime }}_{1}$, ${y^{\prime }}_{1}$, ${z^{\prime }}_{1}$ as initial values to avoid the transient effect; then to obtain three chaotic sequences from the *N*_0_ + 1-th value, (${x^{\prime }}_{i},i=1,2,\ldots$), (${y^{\prime }}_{i},i=1,2,\ldots$), (${z^{\prime }}_{i},i=1,2,\ldots$).

**Step****3:** An integer sequence $t_{i}$ ranging from [0,2] is calculated by using Equation (7):(7)$$t_{i}=\mathrm{floor}({x^{\prime }}_{i}\times 10^{3})\mathrm{mod}2+1$$

**Step****4:** Substitute $y_{1}^{i}$, and $z_{1}^{i}$ into Equations (8) and (9) to obtain an integer sequence *Y_i_* and *Z_i_* ranging from [0,255], respectively.
(8)$$Y_{i}=\mathrm{floor}({y^{\prime }}_{i}\times 10^{3})\mathrm{mod}2^{8}$$
(9)$$Z_{i}=\mathrm{floor}({z^{\prime }}_{i}\times 10^{3})\mathrm{mod}2^{8}$$

**Step 5:** The array *S* can be calculated by the following Equation (10). The number stored in *S* is not repeated. When array *S* is filled, the S-box is obtained.
(10)$$S(i)=\left\{ \begin{array}{l} Y_{i},\;t_{i}=1\\ Z_{i},\;t_{i}=2 \end{array}\right.$$

**Step****6:** The integer sequence *S* is arranged into a 16 × 16 table to obtain an initial prototype S-box. 

**Step****7:** Perform the stretch and fold transform on the initial prototype S-box (see Figure 4a–c, and turn upside down, perform the stretch and fold transform for *M*_0_ times on the transformed S-box to obtain the proposed S-box (see Table 1).

### 3.2. Performance Analysis of the Designed S-Box

We compare our obtained S-box with many S-boxes generated in [33,34,35,36,37,38,39,40,41,42,43,44,45,46,47,48,49,50,51,52,53,54,55]. Initialization parameters are configured as the following: $x_{0}$ = 0.36, $y_{0}$ = 0.25, $z_{0}$ = 0.78, 𝑘_1_ = 32.5, 𝑘_2_ = 34.3, 𝑘_3_ = 28.1, 𝜇 = 3.999, $N_{0}$ = 1000, and $M_{0}$ = 757. The obtained S-box is described in Table 1, and Table 2 lists the comparison results.

We select five criteria including nonlinearity [71], strict avalanche criterion (SAC) [72], bits independence criterion (BIC) [73], differential approximation probability (DP) [74], and linear approximation probability (LP) [75] to verify the cryptographic characteristics of our S-box.

The minimum, maximum, and average nonlinearities of the obtained S-box are 104, 110, and 107, respectively, as shown in Table 2. Thus, the obtained S-box is resistant to differential linear attacks.

The dependency matrix of the obtained S-box is shown in Table 2. Specifically, the achieved maximum, minimum, and average values of the obtained S-box are 0.5781, 0.4219, and 0.4954, respectively. The following observations can be obtained from Table 2: (1) the S-boxes designed in refs. [34,36,38,40,44,45,46,47,48,51,52,53,55] outperform the obtained S-box; (2) however, the average value of the obtained S-box is closer to the ideal value, 0.5, than those designed in refs. [33,35,37,39,40,41,43,49,50,54]. Therefore, the designed S-box meets the SAC criterion.

The BIC test results of the obtained S-box are depicted in Table 2. In the table, the average BIC-nonlinearity and the average BIC-SAC value of the obtained S-box are 102.93 and 0.5034, respectively. The following conclusion can be obtained from this table: (1) the average BIC-nonlinearity value of the obtained S-box is lower than those designed in refs. [33,34,36,37,39,41,44,45,46,47,48,49,50,51,52,53,54,55]; (2) the average BIC-nonlinearity value of the obtained S-box is greater than those constructed in refs. [35,38,40,43]; (3) the average BIC-SAC value of the obtained S-box is better than those proposed in refs. [34,35,40,44,45,46,49,50,53], which is closer to the ideal value of 0.5. Therefore, the obtained S-box can meet the BIC characteristic.

The DP test results of all comparison S-boxes are depicted in Table 2. From this table, we can draw the following observations: (1) those constructed S-boxes in refs. [34,35,37,38,41,42,44,46,47,48,49,51,52,53,54,55] can obtain larger DP values than the obtained S-box; (2) yet, the DP value of those of the other comparison S-boxes is not better than the obtained S-box. Thus, the obtained S-box can resist a differential cryptanalysis attack effectively.

Additionally, the maximum LP value of the obtained S-box is 0.148438 in Table 2. From these results, we can make the following observations: (1) the LP value of the generated S-boxes in refs. [33,34,35,36,37,39,40,41,42,44,45,46,47,48,49,50,51,52,53,54,55] are better than the obtained S-box; (2) the LP value of the obtained S-box is not worse than those of the other comparison S-boxes. Therefore, the obtained S-box can also resist the linear cryptanalysis attack effectively.

## 4. Proposed Encryption and Decryption Framework

### Encryption Scheme

The encryption scheme is presented in Figure 5. We suppose the grey plaintext image *P* is of size M×N, and the pixel value ranges from 0 to 255. The detailed description is presented as below.

**Step****1:** Substituting the initial values $x_{0}$, $y_{0}$, $z_{0}$ into Equation (3), we can obtain $x_{1}$, $y_{1}$, $z_{1}$ after *N*_0_ iterations.

**Step****2:** Consider $x_{1}$, $y_{1}$, $z_{1}$ as initial values; then to obtain three chaotic sequences from the *N*_0_ + 1-th value, ($x_{i},i=1,2,\ldots ,M\times N$), ($y_{i},i=1,2,\ldots ,M\times N$), ($z_{i},i=1,2,\ldots ,M\times N$).

**Step****3:** Substitute $x_{i}$, $y_{i}$ and $z_{i}$ into Equations (11)–(13) to obtain an integer sequence $xx_{i}$, $yy_{i}$ and $zz_{i}$ ranging from [0,255], respectively.
(11)$$xx_{i}=floor(x_{i}\times 10^{3})\mathrm{mod}2^{8}\;i=1,2,\ldots ,M\times N$$
(12)$$yy_{i}=floor(y_{i}\times 10^{3})\mathrm{mod}2^{8}\;i=1,2,\ldots ,M\times N$$
(13)$$zz_{i}=floor(z_{i}\times 10^{3})\mathrm{mod}2^{8}\;i=1,2,\ldots ,M\times N$$

**Step 4:** For the plain image *P*, use a binary value *w* = *b*_7_ *b*_6_
*b*_5_ *b*_4_ *b*_3_
*b*_2_
*b*_1_ *b*_0_ to represent a pixel of it. Denote *i* = *b*_7_ *b*_6_ *b*_5_ *b*_4_ by a binary representation of a row index value, and convert it to a decimal value *s*. Denote *j* = *b*_3_ *b*_2_ *b*_1_ *b*_0_ by a binary representation of a column index value, and convert into a decimal value *t*. Replace the elements of the *s* +1th row of the S-box and that of the *t* +1th column to obtain the plain image $P^{\prime }$.

**Step****5**: Arrange the pixels of $P^{\prime }$ into a 1-D sequence *B =* {*b*_1_, *b*_2_,…, *b_M_*_×*N*_} in left-to-right and top-to-bottom order.

**Step 6:** Calculate $c_{i}$ by Equation (14):(14)$$c_{i}=b_{i}⊕zz_{i}$$
where ⊕ is the XOR operation, and $c_{i}$ is the output pixel data.

**Step 7:***m*(*x*, *y*) is formed by the matrix {$xx_{i}$′ (*i*, *j*)|*i* = 1, 2,…,*M*; *j* = 1, 2,…,*N*} after rearranging the sequence $xx_{i}$ {i=1,2,…,M×N}. Similarly, *n*(*x*, *y*) is produced by the matrix {$yy_{i}$′ (*i*, *j*)|*i* = 1, 2,…,*M*; *j* = 1, 2,…,*N*} after rearranging the sequence $yy_{i}$ {i=1,2,…,M×N}.

**Step 8:** Perform FT and IFT according to Equation (1), and then to obtain an image *D*.

**Step 9:** The average value of the plaintext *P* is calculated by Equation (15):(15)$$mean=\frac{\sum_{x\in [1,M],\;y\in [1,N]}P(x,y)}{M\times N}$$

**Step 10:** The parameter *k* is calculated by Equation (16):(16)$$k=\mathrm{mod}\left(\left\lfloor mean\times 10^{5}+x_{0}\times 10^{8}+y_{0}\times 10^{8}+z_{0}\times 10^{8}\right\rfloor ,256\right)$$

**Step****11:** Substitute *k* into Equation (6); that is IAT. Then scramble the complex-valued image *D*; finally, encrypted image *E* can be obtained. It is noteworthy that the encrypted image *E* is a complex-valued distribution. 

In our paper, the proposed scheme is symmetric. The decryption process will not be repeated here.

## 5. Simulation and Security Analysis 

### 5.1. Simulation Results

The test images include “Lena”, “Boat”, “Cameraman”, “Peppers”, “House”, “Lake”, “Moon surface”, and “Plane”, and the size of eight gray images is 256 × 256 (from USC-SIPI Image Database [76]). The keys are as follows:$x_{0}$ = 0.16, $y_{0}$ = 0.36, $z_{0}$ = 0.56, 𝑘_1_ = 32.5, 𝑘_2_ = 34.3, 𝑘_3_ = 28.1, 𝜇 = 3.999, $N_{0}$ = 1000, h = 2, g = 1, e = 1, r = 3. All experiments are conducted in MATLAB R2012b. The operating system is Microsoft Windows 7. The hardware environment is a 2.9 GHz CPU and 16 GB memory. The simulation results of the proposed scheme on the above four gray images are shows in Figure 6.

### 5.2. Security Analysis

#### 5.2.1. Key Space Analysis

In the proposed scheme, the initial values $x_{0}$, $y_{0}$, $z_{0}$ and the parameters 𝑘_1_, 𝑘_2_, 𝑘_3_, 𝜇 of a chaotic system are used as the secret keys. If the precision is set to 10^−14^, the key space of this scheme is 10^14^ × 7 = 10^112^ ≈ 2^372^ >> 2^100^ [77], which can resist brute-force attacks effectively.

#### 5.2.2. Key Sensitivity Analysis

The key is sensitive if a slight change in the key with other keys remaining unchanged will result in a completely different decryption result. Figure 7 shows sensitivity analysis results of security keys. Figure 7a–c show the decrypted images corresponding to the encrypted image Figure 6b with the keys $x_{0}$ = 0.16 + 10^−14^, *k*_1_ = 32.5 + 10^−14^, 𝜇 = 3.999 + 10^−14^, respectively. Therefore, the proposed algorithm scheme exhibits high key sensitivity.

#### 5.2.3. Histogram Analysis 

The histogram of an excellent encryption image should be uniformly distributed [56]. The histograms of the plaintext images are shown in Figure 8a,c,e,g,i,k,m and o, respectively. The histograms of the corresponding encrypted images are shown in Figure 8b,d,f,h,j,l,n and p, respectively. 

We can observe that the image information distribution of ciphertext encrypted with different plaintext is uniform. Therefore, the scheme proposed can resist statistical attacks.

#### 5.2.4. Chi-Square Test Analysis 

To analyze the distribution of the encrypted image histogram intuitively, we perform a Chi-square test. The more uniform the encrypted image pixels are, the lower the Chi-square value is.

The encrypted Chi-square test can be calculated by Equation (17) [56]:(17)$$\chi^{2}=\sum_{i=0}^{255}\frac{\left(q_{i}-q\right)}{q},\;q=\frac{M\times N}{256}$$
where *q*_i_ is the number of pixels *i*, and *M* × *N* the size of a cipher image.

The Chi-square test results of the encrypted images are given in Table 3. At the 5% significance level, the Chi-square value $\chi_{0.05}^{2}$ = 293.2478. Table 3 shows that the test results of cipher images are not greater than 293.2478. Thus, the proposed scheme has the ability of resisting statistical attacks.

#### 5.2.5. Mean Squared Error and Peak Signal-To-Noise Ratio Analysis 

The mean square error (MSE) and Peak Signal-to-Noise Ratio (PSNR) are explored to assess the difference between two images [78].

The MSE and PSNR can be computed by Equations (18) and (19), respectively:(18)$$\mathrm{MSE}=\frac{\sum_{i,j}{\left(C(i,j)-D(i,j)\right)}^{2}}{M\times N}$$
(19)$$\mathrm{PSNR}=10\log_{10}(\frac{255^{2}}{\mathrm{MSE}})$$
where *M* × *N* is the size of an image, and *C*(·) and *D*(·) are the corresponding pixel values of two comparison images. The smaller the PSNR between the original image and the encrypted one, the better the encryption effect. If the PSNR value equals infinity, then the two images are the same. The test results in Table 4 show that the proposed scheme can achieve a good encryption effect.

#### 5.2.6. Correlation Analysis

To resist statistical analysis attacks, the encryption scheme should reduce the correlation of adjacent pixels of the encrypted image.

We calculate the correlation coefficients of the plain image and the cipher image according to Equations (20)–(23) [57]:(20)$$\gamma_{xy}=\frac{\mathrm{cov}(x,y)}{\sqrt{D(x)}\sqrt{D(y)}}$$
(21)$$\mathrm{cov}(x,y)=E((x_{i}-E(x))(y_{i}-E(y)))$$
(22)$$E(x)=\frac{1}{N}\sum_{i=1}^{N}x_{i}$$
(23)$$D(x)=\frac{1}{N}\sum_{i=1}^{N}{\left(x_{i}-E(x)\right)}^{2}$$

In this paper, we randomly chose 3000 pairs of adjacent pixels from a plaintext image “Lena” and its corresponding ciphertext image, and calculated their correction coefficients in horizontal, vertical, and diagonal directions. The adjacent pixel correlation distributions of Lena’s plaintext image in each direction are shown in Figure 9a–c, respectively. The adjacent pixel correlation distributions of the magnitude of Lena’s encrypted image in each direction are shown in Figure 9d–f, respectively. The correlation coefficients of Lena in each direction are shown in Table 5. The comparison results also reflect that our scheme indicates a negligible correlation. Thus, our scheme can resist statistical attacks effectively.

#### 5.2.7. Differential Attack Analysis

The number of pixels change rate (NPCR) and unified average changing intensity (UACI) can be utilized to analyze the differential attack performance of an encryption algorithm scheme. NPCR is often utilized to measure the absolute number of value-changed pixels in differential attacks, and UACI is often utilized to measure the averaged difference between two paired ciphertext images. They are computed by Equations (24) and (25), respectively [58]:(24)$$\mathrm{NPCR}=\frac{\sum_{i,j}D(i,j)\times 100\%}{M\times N}$$
(25)$$\mathrm{UACI}=\frac{\sum_{i,j}\frac{\left|C_{1}(i,j)-C_{2}(i,j)\right|}{255}\times 100\%}{M\times N}$$
where $C_{1}(i,j)$, $C_{2}(i,j)$ denote two different encrypted images. *M* × *N* represents the size of an image. If $C_{1}(i,j)\neq C_{2}(i,j)$, then D(i,j)=1, otherwise, D(i,j)=0.

The ideal value of NPCR is 99.6093%, and the ideal value of UACI is 33.4635% [61]. The NPCR and UACI values of ciphertexts of different images obtained by our scheme are shown in Table 6. The simulation results show that the NPCR and UACI values of the proposed scheme are close to the ideal values. Therefore, our scheme can resist differential attacks effectively.

#### 5.2.8. Robustness Analysis

We utilize data loss attacks and noise attacks to evaluate our scheme robustness. 

a.Data loss attack

To evaluate the performance of our scheme in resisting data loss attacks [60], the encryption images with 1/16, 1/8, 1/4, and 1/2 data loss are shown in Figure 10a,c,e and g, and the corresponding decryption images are shown in Figure 10b,d,f and h, respectively. The simulation results show that our proposed scheme can resist loss attacks effectively in both Table 7 and Figure 10.

b.Noise attack

To evaluate the scheme performance, we added salt-and-pepper noise with different intensities and Gauss noise with diverse variances to a cipher image and then decrypted the noise-added cipher images [62]. The Gauss noise variances were 0.2, 0.3, 0.4 and 0.5, and the salt-and-pepper noise intensities were 0.001, 0.01, 0.05, and 0.1. In Table 6, the noise intensity and the values of the corresponding PSNR and MES are shown. Obviously, the proposed scheme can resist noise attacks effectively, as shown in both Table 8 and Figure 11.

#### 5.2.9. Entropy Analysis

The information entropy is used to reflect the randomness of the result of the encrypted image. The information source is denoted by *t*, and the entropy value is calculated by Equation (26) [63]:(26)$$H(t)=\sum_{i=0}^{255}p(t_{i})\log_{2}\frac{1}{p(t_{i})}$$
where $p(t_{i})$ is the probability of the occurrence of pixel gray value *t_i_*.

The closer it is to 8, the more disordered the information. Table 9 lists the entropy values of ciphertexts of different images encrypted by our scheme. The entropy values of the encrypted Lena image by the proposed scheme in refs. [56,58,59] is depicted in Table 10. The entropy value of our scheme is closer to 8, as shown in Table 9 and Table 10, which demonstrates that the scheme proposed is effective.

#### 5.2.10. Speed Analysis

The simulation was conducted in MATLAB R2012b on a computer with the operating system Windows 7 and 16-GB RAM, Intel(R) Core(TM) i5-9400 CPU @ 2.90 GHz. The average running time of this encryption/decryption scheme for encrypting/decrypting the Lena image was 3.081 s.

## 6. Conclusions

In our paper, a novel virtual optical image encryption scheme combining a chaotic S-box, DRPE, and IAT was proposed. In the proposed scheme, a new chaotic S-box was constructed. We XORed the substituted image with a nonlinear chaotic sequence. The XOR result was encoded by DRPE, and the encoded result was further scrambled by IAT. Simulation results show that our scheme can resist statistical attacks, brute-force attacks, and differential attacks.

## Figures and Tables

**Figure 1 sensors-22-05325-f001:**
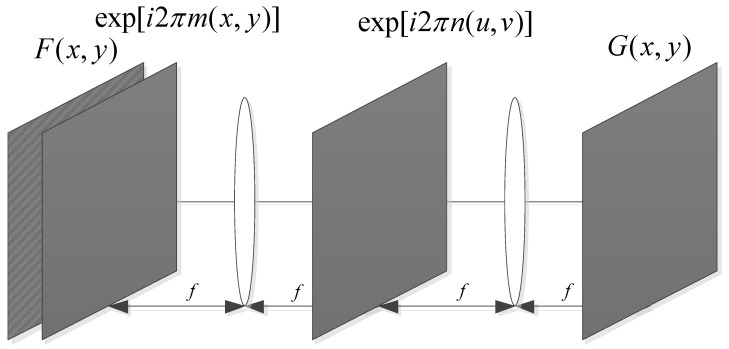
The architecture of the optical DRPE cryptosystem.

**Figure 2 sensors-22-05325-f002:**
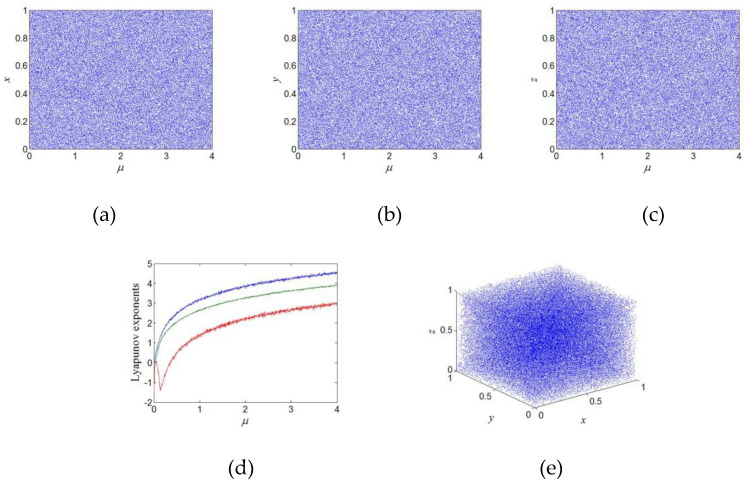
Chaotic bifurcation diagram, Lyapunov exponents, and attractor diagram: (**a**) the chaotic bifurcation diagram of the chaotic map x; (**b**) the chaotic bifurcation diagram of the chaotic map y; (**c**) the chaotic bifurcation diagram of the chaotic map z; (**d**) Lyapunov exponents; (**e**) the chaotic attractor in the x-y-z plane.

**Figure 3 sensors-22-05325-f003:**
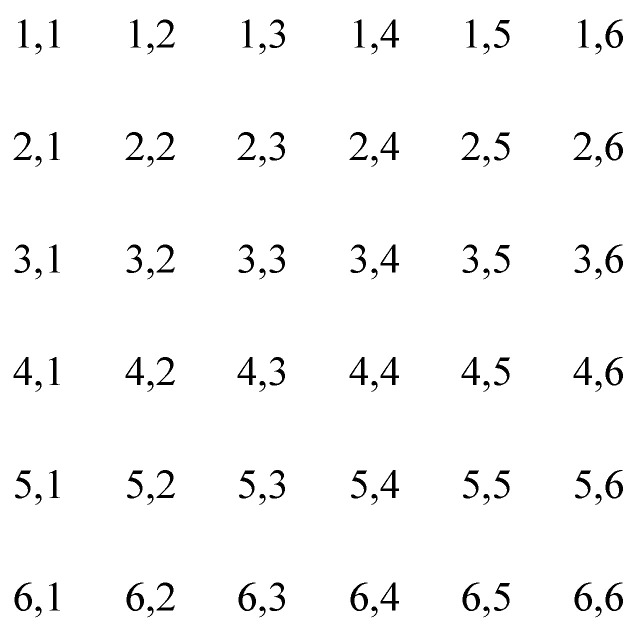
A 6 × 6 image matrix.

**Figure 4 sensors-22-05325-f004:**
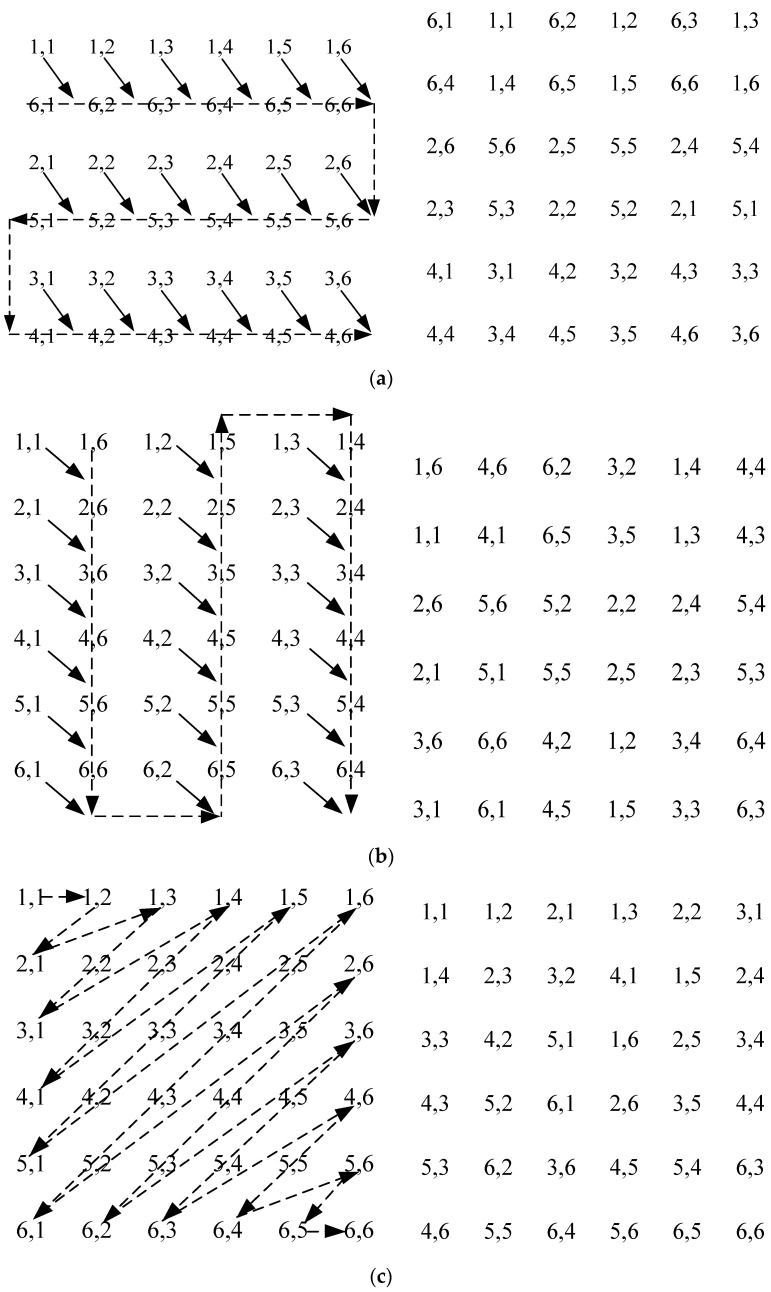
Process of the transform. (**a**) The stretch transform of nonadjacent rows; (**b**) The stretch transform of nonadjacent columns; (**c**) the fold transform of a snake line.

**Figure 5 sensors-22-05325-f005:**
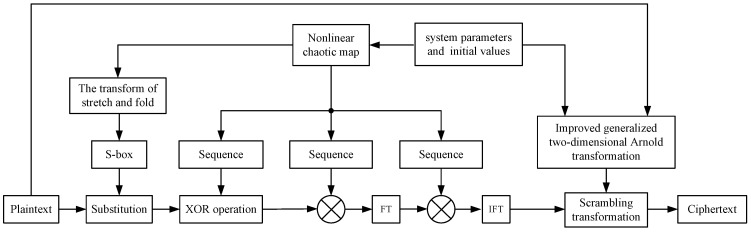
The architecture of the developed encryption scheme.

**Figure 6 sensors-22-05325-f006:**
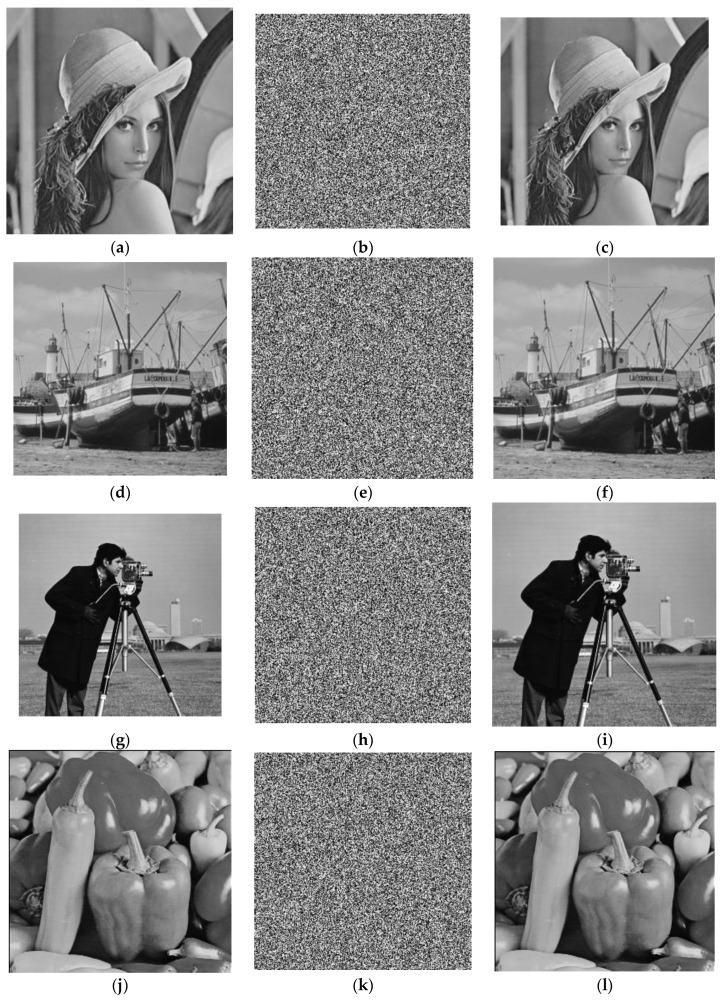
Simulation results of four gray images. (**a**) plaintext image of Lena; (**b**) the amplitude of encrypted image of Lena; (**c**) decrypted image of Lena; (**d**) plaintext image of Boat; (**e**) the amplitude of encrypted image of Boat; (**f**) decrypted image of Boat; (**g**) plaintext image of Cameraman; (**h**) the amplitude of encrypted image of Cameraman; (**i**) decrypted image of Cameraman; (**j**) plaintext image of Pepper; (**k**) the amplitude of encrypted image of Pepper; (**l**) decrypted image of Pepper; (**m**) plaintext image of House; (**n**) the amplitude of encrypted image of House; (**o**) decrypted image of House; (**p**) plaintext image of Lake; (**q**) the amplitude of encrypted image of Lake; (**r**) decrypted image of Lake; (**s**) plaintext image of Moon surface; (**t**) the amplitude of encrypted image of Moon surface; (**u**) decrypted image of Moon surface; (**v**) plaintext image of Plane; (**w**) the amplitude of encrypted image of Plane; (**x**) decrypted image of Plane.

**Figure 7 sensors-22-05325-f007:**
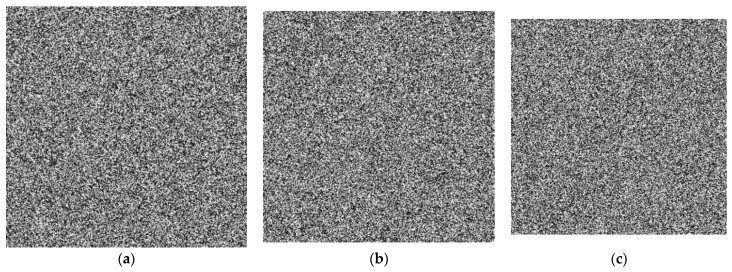
Decrypted image using incorrect keys (**a**)$x_{0}$ = 0.16 + 10^−14^; (**b**) *k*_1_ = 32.5 + 10^−14^; (**c**) 𝜇 = 3.999 + 10^−14^.

**Figure 8 sensors-22-05325-f008:**
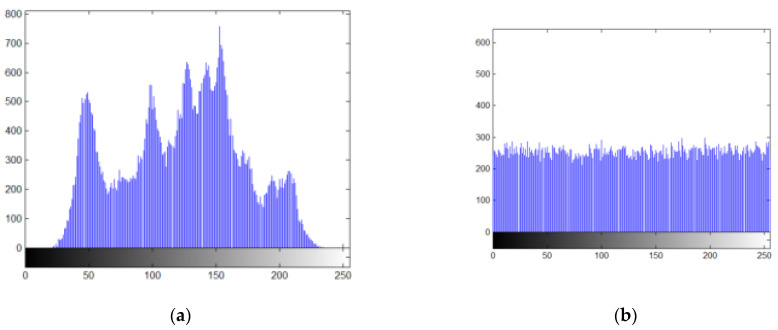
Histograms of the magnitude of the plaintext and encrypted images. (**a**) Lena; (**b**) ciphered Lena; (**c**) Boat; (**d**) ciphered Boat; (**e**) Cameraman; (**f**) ciphered Cameraman; (**g**) Peppers; (**h**) ciphered Peppers; (**i**) House; (**j**) ciphered House; (**k**) Lake; (**l**) ciphered Lake; (**m**) Moon surface; (**n**) ciphered Moon surface; (**o**) Plane; (**p**) ciphered Plane.

**Figure 9 sensors-22-05325-f009:**
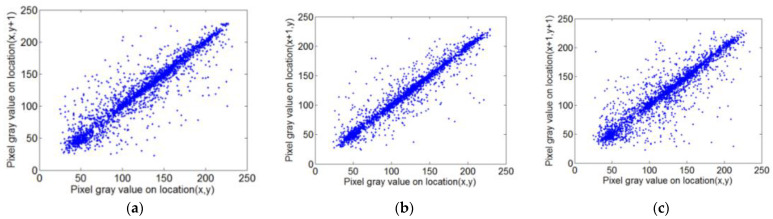
Correlation coefficients of Lena. (**a**) Horizontal correlation of plain image; (**b**) vertical correlation of plain image; (**c**) diagonal correlation of plain image; (**d**) horizontal correlation of the magnitude of ciphered image; (**e**) vertical correlation of the magnitude of ciphered image; (**f**) diagonal correlation of the magnitude of ciphered image.

**Figure 10 sensors-22-05325-f010:**
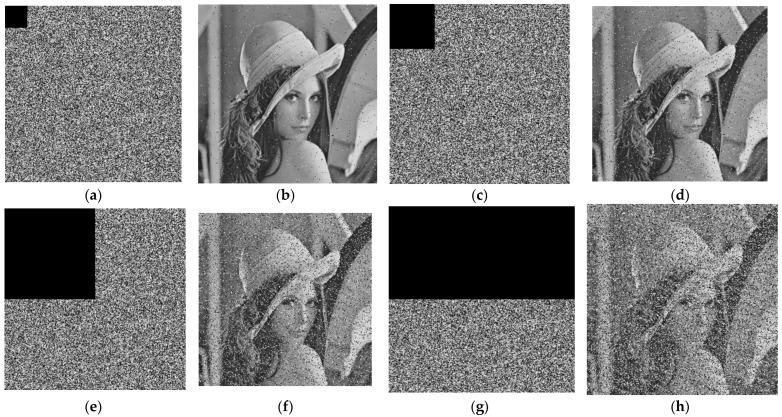
Data loss attack test. (**a**) 1/16 cropped; (**b**) corresponding decrypted image of (**a**); (**c**) 1/8 cropped; (**d**) corresponding decrypted image of (**c**); (**e**) 1/4 cropped; (**f**) corresponding decrypted image of (**e**); (**g**) 1/2 cropped; and (**h**) corresponding decrypted image of (**g**).

**Figure 11 sensors-22-05325-f011:**
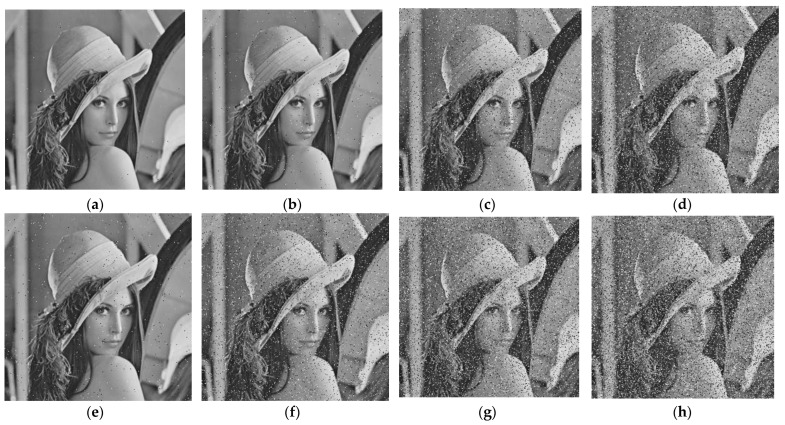
Noise attack test. (**a**–**d**) with salt-and-pepper noise densities of 0.001, 0.01, 0.05, 0.1, respectively; (**e**–**h**) with Gaussian white noise variance values of 0.2, 0.3, 0.4,0.5 respectively.

**Table 1 sensors-22-05325-t001:** The obtained S-box.

No.	1	2	3	4	5	6	7	8	9	10	11	12	13	14	15	16
1	132	211	219	33	46	9	224	202	155	187	34	154	13	28	102	221
2	117	228	166	250	63	159	177	100	182	58	85	170	60	52	238	64
3	142	6	78	247	248	4	108	18	68	107	194	45	209	50	87	119
4	158	54	37	109	150	114	65	181	234	14	243	123	76	21	72	217
5	3	24	227	113	0	19	208	17	111	70	171	110	156	2	145	152
6	5	192	231	89	193	240	244	20	215	149	173	229	180	40	255	49
7	201	143	179	169	147	32	137	47	15	239	176	253	80	252	204	225
8	44	165	73	105	56	160	133	134	191	55	206	1	183	12	203	36
9	35	7	74	184	212	129	8	23	26	127	122	162	172	242	118	214
10	223	120	16	125	207	199	148	226	144	95	51	71	103	41	77	178
11	88	222	174	164	146	130	188	126	216	81	200	249	29	140	157	10
12	189	91	22	98	198	205	61	161	190	151	94	245	233	163	195	136
13	25	31	97	79	39	82	135	218	141	11	196	168	186	175	101	121
14	69	27	30	115	53	42	210	246	220	232	96	116	90	43	83	237
15	48	139	241	213	92	106	59	124	153	86	197	138	112	93	67	254
16	75	38	84	57	104	251	236	131	66	235	167	230	99	185	128	62

**Table 2 sensors-22-05325-t002:** Cryptanalysis comparison results of S-boxes.

S-Boxes	Nonlinearity			SAC			BIC- Nonlinearity	BIC- SAC	DP	LP
	Min	Max	Avg.	Min	Max	Avg.				
Proposed	104	110	107	0.4219	0.5781	0.4954	102.93	0.5034	0.04688	0.148438
Ref. [33]	100	108	103.25	0.3750	0.5938	0.5059	104.29	0.5031	0.04688	0.125000
Ref. [34]	104	108	105	0.4063	0.5781	0.4971	103	0.5044	0.03906	0.132813
Ref. [35]	101	108	103.88	0.3906	0.5781	0.5059	102.68	0.4958	0.03906	0.132813
Ref. [36]	100	106	103	0.4219	0.6094	0.5000	103.14	0.5024	0.05469	0.132813
Ref. [37]	100	106	103.25	0.4219	0.5938	0.5049	103.71	0.5010	0.03906	0.132813
Ref. [38]	96	106	103	0.3906	0.6250	0.5039	100.36	0.5010	0.03906	0.148438
Ref. [39]	102	108	104.75	0.3906	0.5938	0.5056	104.07	0.5022	0.04688	0.125000
Ref. [40]	98	108	104.25	0.2813	0.6094	0.4954	102.86	0.5048	0.04688	0.140625
Ref. [41]	100	106	104	0.3750	0.6250	0.4946	103.21	0.5019	0.03906	0.132813
Ref. [42]	100	106	103	0.3906	0.5938	0.5020	102.93	0.4999	0.03906	0.140625
Ref. [43]	84	106	100	0.1250	0.6250	0.4812	101.93	0.4967	0.06250	0.179688
Ref. [44]	104	110	106.25	0.4219	0.5938	0.5039	103.36	0.5059	0.03906	0.140625
Ref. [45]	100	110	105.50	0.4063	0.6094	0.5010	103.79	0.5036	0.04688	0.132813
Ref. [46]	101	107	104.5	0.4219	0.5781	0.4963	103.29	0.4938	0.03906	0.140625
Ref. [47]	104	108	106.75	0.4063	0.6250	0.4976	103.57	0.5022	0.03906	0.132813
Ref. [48]	104	108	106.25	0.3594	0.6094	0.5002	103.64	0.4993	0.03906	0.132813
Ref. [49]	112	112	112	0.4531	0.5625	0.5051	112	0.5044	0.01560	0.062500
Ref. [50]	102	108	105.25	0.4688	0.5938	0.5352	103.21	0.5085	0.05469	0.140625
Ref. [51]	106	110	108.5	0.4063	0.5781	0.4995	103.86	0.5016	0.03906	0.132813
Ref. [52]	110	112	112.5	0.4063	0.5938	0.4985	103.79	0.5014	0.03906	0.132813
Ref. [53]	104	110	107	0.4219	0.5938	0.4993	103.29	0.5051	0.03906	0.132813
Ref. [54]	102	108	105.5	0.4219	0.5781	0.5061	103	0.5009	0.03906	0.140625
Ref. [55]	102	110	106.5	0.4063	0.5938	0.5010	103.43	0.4980	0.0391	0.132813

**Table 3 sensors-22-05325-t003:** Chi-square test results of encrypted images.

Cipher Image	Chi-Square Value	Result
Lena	246.4219	Pass
Boat	242.7656	Pass
Cameraman	244.4688	Pass
Peppers	248.6797	Pass
House	262.9219	Pass
Lake	240.4375	Pass
Moon surface	287.6328	Pass
Plane	240.6016	Pass

**Table 4 sensors-22-05325-t004:** Test results of the encryption and decryption performances by different metrics.

Test Image	MSE (Original vs. Encrypted)	PSNR (Original vs. Encrypted)	MSE (Original vs. Decrypted)	PSNR (Original vs. Decrypted)
Lena	7802.8866	9.2083	0	∞
Boat	8263.2444	8.9593	0	∞
Cameraman	9439.7874	8.3812	0	∞
Peppers	8193.0659	8.9963	0	∞
House	8454.3259	8.8600	0	∞
Lake	10,728.4255	7.8254	0	∞
Moon surface	6217.3002	10.1948	0	∞
Plane	8987.5783	8.5944	0	∞

“∞” denotes “infinity” in the related study.

**Table 5 sensors-22-05325-t005:** Correlation coefficients of Lena.

Test Image	Horizontal	Vertical	Diagonal
Plaintext image Lena	0.9051	0.9652	0.9293
Encrypted image Lena	−0.0053	−0.0012	0.0050
Ref. [56]	0.0032	−0.0003	0.0012
Ref. [58]	−0.0056	0.0006	0.0018
Ref. [59]	−0.0009	−0.0030	0.0062

**Table 6 sensors-22-05325-t006:** The NPCR and UACI values of encrypted images.

Test Image	NPCR (%)	UACI (%)
Lena	99.6216	33.6642
Boat	99.6017	33.3610
Cameraman	99.5987	33.3694
Peppers	99.6033	33.5057
House	99.5865	33.3840
Lake	99.5987	33.4928
Moon surface	99.5880	33.3971
Plane	99.5941	33.4519

**Table 7 sensors-22-05325-t007:** PSNR and MES testing results subject to data loss attacks.

Data Loss	1/16	1/8	1/4	1/2
PSNR	9.0229	8.7816	7.6823	6.3940
MSE	8143.1129	8608.3023	11,087.8870	14,917.0311

**Table 8 sensors-22-05325-t008:** PSNR and MES testing results under noise attacks.

Noise Type	Salt-and-Pepper Noise				Gaussian White Noise			
Noise intensity	0.001	0.01	0.05	0.1	0.2	0.3	0.4	0.5
PSNR	9.2002	9.1446	8.4837	9.2083	9.1789	9.0572	8.8713	8.6525
MSE	7817.4412	7918.1615	9219.6355	7802.8866	7855.7743	8079.0790	8432.4675	8868.1909

**Table 9 sensors-22-05325-t009:** Entropy values of different images.

Test Image	Plaintext Image	Encrypted Image
Lena	7.4551	7.997286
Boat	7.1011	7.997322
Cameraman	7.0097	7.997325
Peppers	7.5251	7.997258
House	6.5637	7.997106
Lake	7.3767	7.997353
Moon surface	6.7093	7.996827
Plane	6.9860	7.997342

**Table 10 sensors-22-05325-t010:** Entropy values of the Lena image encrypted by different comparison schemes.

Encryption Schemes	Entropy Values
Proposed	7.9973
Ref. [56]	7.9994
Ref. [58]	7.9971
Ref. [59]	7.9973

## Data Availability

Data are available within the manuscript.

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
