# Peer review of "A Novel Virtual Optical Image Encryption Scheme Created by Combining Chaotic S-Box with Double Random Phase Encoding"

_sensors, 2022, doi:10.3390/s22145325_

Round 1

Reviewer 1 Report

Title: A novel virtual optical image encryption scheme by combining chaotic S-box with double random phase encoding.

Authors: Peiran Tian, Runzhou Su.
Manuscript ID: sensors-1791093.

This paper presents a virtual optical image encryption based on substitution box, nonlinear chaotic maps, improved Arnold transform and the double random phase encoding (DRPE). The sections of the paper are not well written in English and there are some mistakes in some equations, functions or parameters. Several sections of the paper should be improved and explained in more detail. Finally, some results need to be improved in their presentation and some other metrics should be computed.

I consider that this paper should have major changes before to publish it in this journal due to the following reasons:

1. The redaction of the paper in English need to be improved. In several parts of the paper, the writing is sloppy and there are many typos.

2. Please, change the word leverage for other word more suitable. 

3. The references [1] and [2] are not appropriate to the text mentioned in the section 1. Introduction because these references are not seminal or foundational paper about optical encryption.

4. In section 1. Introduction, what is the meaning of IAF?

5. At the end of section 1, use the word Section instead of Sect.

6. The section 2.1 Overview of optical DRPE cryptosystem has to be properly explained, because the random phase masks (RPMs) of the DPRE are not used to be scrambled, the second RPM is not mentioned in this section and the reference [39] is not suitable for this section.

7. In section 2.2, the parameters k should have subindices 1, 2 and 3.

8. In section 3.1, Step 3, the words using (7)” should be deleted. What is the motivation of equation (7), if afterwards it is not used at all?

9. In section 3.2, the parameters k with subindices 1, 2 and 3 are not defined.

10. The results of Table 2 should be presented in a more suitable way.

11. Section 4.1, the Step 4 is very poorly written and hard to understand, please, rewrite it.

12. Section 4.1, the Step 7 is wrong because the images to be formed are represented by m(x, y) and n(u, v) according to the equation (1) of this paper.

13. Section 4.1, the Step 8 is wrong because the only equation to be used in encryption scheme is the equation (1). The equation (2) is for decrypting scheme.

14. Section 4.1, the Arnold transform has to be mentioned in Step 11. In this step of encryption scheme, the authors have to mention that the encrypted image is a complex-valued distribution. Therefore, the encrypted image has magnitude and phase.

15. Again, the parameters k with subindices 1, 2 and 3 are not defined in section 5.1. When the parameters of the encryption are mentioned in this section, the authors should mention the mathematical operation for each parameter.

16. In section 5.1, some metric for measure the quality of the decrypted image need to be computed and analyzed.

17. The images of figure 6(b), (d), (f) and (h) represent the magnitude or the phase of the encrypted image?

18. Again, the parameters k with subindices 1, 2 and 3 are not defined in section 5.2.1. In this subsection, what is the meaning of violent attacks? I guess the proper phrase is brute-force attacks.

19. The histograms of figure 8(b), (e), (h) and (k) represent the magnitude or the phase of the encrypted image?

20. Please, write the meaning of NPCR and UACI in section 5.2.5. In this section, authors should use MxN instead of WxH.

21. In section 5.2.6, the metric of MSE is wrong because this metric must be defined in this section in order to measure the quality of the decrypted images. Again, authors should use MxN instead of WxH.

22. The section 6 should be improved according to the new modifications of the paper. In this section, when the authors mention: “The positions of pixels were permuted by the IAT”, which image is scrambled?

23. Finally, when the authors mention: “… satisfies the cryptographic requirement” these words should be used carefully because the authors cannot demonstrate the resistance of the proposed encryption scheme against all possible attacks.

Author Response

Dear reviewer,

Please refer to the attached PDF file named 'sensors-1791093-coverletter' for our response to your comments. Thank you for the time and effort spent in reviewing this paper and for your consideration of the improved version. 

Sincerely, The authors.

Reviewer 2 Report

In this manuscript, the authors present an optical image encryption method mainly using double random phase encoding. There are several serious concerns as follows:

1) In recent years, there are many research work to develop optical methods for image encryption based on different optical techniques. Compared with other methods, the proposed method is not sufficiently interesting and novel. For example, the main technique, i.e. double random phase encoding, is well known in the research field of the optical information processing community. 

2) What the authors have done in the manuscript is just encrypting an original image using the combination of S-box substitution, double random phase encoding, and Arnold scrambling. The proposed scheme has limited impact either on inspiring the related researchers.

3) It is not surprised that such kind of increment work could be accepted for publication in the early-stage of optical information security fields, but for now there is nothing really new. Additionally, only simulations are given here. It is suggested that experimental work can be conducted to show the validity. 

4) After encryption with double random phase encoding, the ciphertext becomes complex value, which makes the amount of data increase. The overload of the network will be serious when the encrypted data is transmitted.

Author Response

(The authors gave the same response as above.)

Reviewer 3 Report

The paper needs some technical refinements and revisions to make it sound, detailed and suitable for publication. I have the following concerns and suggestions.

1. The authors are advised to clearly mention in the abstract the existing research gap/limitation and how authors tackled them in the paper, mention their contributions, methodology and discuss results superiority in the abstract.

2. Introduction is concisely prepared. Discuss existing issues of the topic and significance & need of the appropriate solutions. Include following findings related to image encryption works.

Efficient modified RC5 based on chaos adapted to image encryption, Journal of Electronic Imaging, 2010, 19(1), 013012

Quantum color image encryption based on multiple discrete chaotic systems, FedCSIS 2017, 2017, pp. 555–559, 8104599

A robust quasi-quantum walks-based steganography protocol for secure transmission of images on cloud-based E-healthcare platforms, Sensors (Switzerland), 2020, 20(11), 3108

Providing End-to-End Security Using Quantum Walks in IoT Networks, IEEE Access, 2020, 8, pp. 92687–92696, 9088144

3. Present exhaustive related survey and mention their features (limitations/demerits). Discuss need of effective optical image enc methods, and sbox methods for usage in efficient image enc.

4. Dedicate a paragraph for sbox description, significance and related sbox methods. Authors have adopted and referred to old sbox methods(papers) they are advised to see some recent sbox publications for sbox related related survey and comparison analysis. authors may go through the following sbox methods.

Bijective S-boxes method using improved chaotic map-based heuristic search and algebraic group structures, IEEE Access, 2020.

A new chaotic substitution box design for block ciphers, SPIN-2014

Comparison of pre and post-action of a finite abelian group over certain nonlinear schemes, IEEE Access, 2020

A novel method of S-box design based on discrete chaotic maps and cuckoo search algorithm, MTAP, 2021.

5. Authors need to provide the Lyapunov spectrum of the chaotic map in eqn(3).

6. Show different image enc statistical performance results for few more images.

7. Include chi-square test analysis.

8. Performance encryption time analysis.

Author Response

(The authors gave the same response as above.)

Round 2

Reviewer 1 Report

Title: A novel virtual optical image encryption scheme by combining chaotic S-box with double random phase encoding.

Authors: Peiran Tian, Runzhou Su.
Manuscript ID: sensors-1791093-v2.

After reviewing the corrections made by the authors, I consider that the following corrections should still be made:

A. The following suggested correction of the first revision was not made in the second version of this paper:

14. Section 4.1, the Arnold transform has to be mentioned in Step 11. In this step of encryption scheme, the authors have to mention that the encrypted image is a complex-valued distribution. Therefore, the encrypted image has magnitude and phase.

B. The following suggested correction of the first revision was not made correctly in the second version of this paper:

21. In section 5.2.6, the metric of MSE is wrong because this metric must be defined in this section in order to measure the quality of the decrypted images. Again, authors should use MxN instead of WxH.

The equation (18) is still wrong because the MSE metric compares a decrypted image with an original image or plaintext as a reference.

C. The correlations of figure 9(d), (e) and (f) are computed using the magnitude or the phase of the encrypted image?

D. The word MES is not correct in Tables 7 and 8, the correct word is MSE.

Author Response

Dear anonymous reviewer,

We are most grateful for your time and effort spent in reviewing the paper. We replied to your comments in a separate response letter named 'author-coverletter-20694390.v1.pdf'. Please refer to the PDF file for our response. 

Sincerely, The Authors.

Reviewer 2 Report

Authors have answered questions well, and the manuscript can be accepted.

Author Response

Dear anonymous reviewer, Thanks for your encouraging comment and your time and efforts spent in reviewing the paper!

Round 3

Reviewer 1 Report

The authors did all the suggested corrections very well.